# Iron Nanoparticles Open Up New Directions for Promoting Healing in Chronic Wounds in the Context of Bacterial Infection

**DOI:** 10.3390/pharmaceutics15092327

**Published:** 2023-09-15

**Authors:** Zhaoyu Lu, Dong Yu, Fengsong Nie, Yang Wang, Yang Chong

**Affiliations:** 1Department of Traditional Chinese Medicine, The Affiliated Hospital of Yangzhou University, Yangzhou University, Yangzhou 225000, China; l17351372862@163.com (Z.L.); ydd0568@163.com (D.Y.); niefengsong@163.com (F.N.); 2Department of General Surgery, The Affiliated Hospital of Yangzhou University, Yangzhou University, Yangzhou 225000, China; 3School of Chemistry and Chemical Engineering, Yangzhou University, Yangzhou 225000, China

**Keywords:** antibacterial ability, chronic wounds, iron nanoparticles, trauma dressings, wound healing

## Abstract

Metal nanoparticles play an outstanding role in the field of wound healing due to their excellent properties, and the significance of iron, one of the most widely used metals globally, cannot be overlooked. The purpose of this review is to determine the importance of iron nanoparticles in wound-healing dressings. Prolonged, poorly healing wounds may induce infections; wound infections are a major cause of chronic wound formation. The primary components of iron nanoparticles are iron oxide nanoparticles, which promote wound healing by being antibacterial, releasing metal ions, and overcoming bacterial resistance. The diameter of iron oxide nanoparticles typically ranges between 1 and 100 nm. Magnetic nanoparticles with a diameter of less than 30 nm are superparamagnetic and are referred to as superparamagnetic iron oxide nanoparticles. This subset of iron oxide nanoparticles can use an external magnetic field for novel functions such as magnetization and functionalization. Iron nanoparticles can serve clinical purposes not only to enhance wound healing through the aforementioned means but also to ameliorate anemia and glucose irregularities, capitalizing on iron’s properties. Iron nanoparticles positively impact the healing process of chronic wounds, potentially extending beyond wound management.

## 1. Introduction

With the growing recognition of nanotechnology, metal nanoparticles (NPs) have progressively entered the domain of wound healing to increase wound recovery from diverse perspectives, including antibacterial properties and the conquering of antibiotic resistance. Concurrently, diverse nanocomposites, such as alginate, collagen, and chitosan, are being incorporated into the wound-healing process. Presently, several wound dressings employing nanoparticles are becoming increasingly accessible and gaining traction in research and clinical settings [1,2]. Given their distinctive physicochemical attributes, iron nanoparticles (IONPs) hold appeal for medical wound dressings due to their remarkable antibacterial and antioxidant features, as well as their impact on biofilms and drug resistance (Figure 1).

In the words of Robert G. Freyberg, chronic trauma, despite molecular-level etiological variances, shares common traits, including elevated pro-inflammatory cytokine levels, proteases, ROS, and senescent cells, alongside persistent infections and deficient stem cell presence [3]. Prolonged subpar healing or failure to heal inevitably leads to a substantial microbial influx in the wound, fostering infection. Conversely, chronic wounds typically stall healing within the inflammatory phase [3,4]. Recognized pathogens, such as staphylococcus aureus, pseudomonas aeruginosa, and beta-hemolytic streptococci, inhabiting chronic wounds contribute to prolonged infections and delayed healing [4,5,6,7,8]. Simultaneously, the considerable number of colonized pathogenic microorganisms within the wound form a persistent aggregated biofilm community. It has been demonstrated that the substantial biofilm present in chronic wounds plays an exceptionally significant role in the progression of wound infection and the delay in healing [9,10,11,12]. Chronic wounds pose a global predicament. Apart from the pain originating from the wounds themselves, they induce various psychological effects and impose an increasing burden on the patient, their family, and the surrounding society. With the escalating trend of an aging population (where wound healing is inversely correlated with age [13]) and the rising costs of global healthcare, coupled with the impact of chronic ailments like diabetes, chronic kidney disease, and obesity, along with a growing awareness of the menace of uncontrollable biofilms and the challenges posed by the recent COVID-19 pandemic, chronic wounds not only perpetuate continuous pain and compromise patients’ quality of life but also evolve into a significant, severe, and all-including clinical, social, and economic issue that no nation can afford to overlook [14,15,16,17,18,19].

**Figure 1 pharmaceutics-15-02327-f001:**
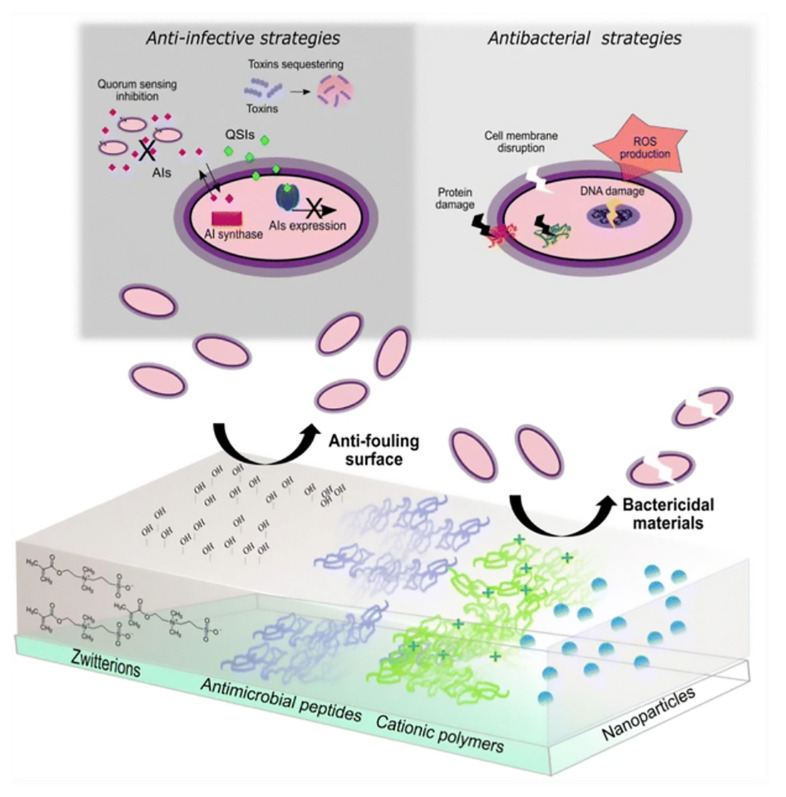
Antiinfection strategies that inhibit the expression of virulence factors and prevent biofilm growth can be categorized as anti-quorum sensing, antitoxin, and anti-biofilm (**left**). Novel antibiotic alternatives aim to mitigate bacterial resistance by eradicating pathogens through nonspecific mechanisms involving cell membrane damage, oxidative stress, and interactions with genetic material and proteins (**right**). Source: Adapted with permission from earlier studies [20], Springer Nature.

Metals have been utilized as antimicrobial agents for an extended period of time. Especially in recent years, antimicrobial metal nanoparticles have assumed a forefront position in infection control due to their distinctive physical and chemical attributes (Figure 2) [21]. In the present era marked by escalating pathogen resistance, although antibiotics remain the primary approach against bacterial infections, the misuse of antibiotics and the emergence of drug-resistant bacteria signify that antibiotics alone cannot entirely address the demands of clinical practitioners and patients (Figure 3). Consequently, the application of metal nanoparticles holds increasing promise (as bacteria generally exhibit reduced resistance to metals) [22,23,24]. The antibacterial efficacy of metals and metal oxide nanoparticles (NPs) has received widespread validation. Combining antibiotics with metal nanoparticles curbs antibiotic consumption while concurrently addressing significant issues like drug resistance and certain adverse reactions, thereby enhancing their bactericidal potency [20,25,26,27]. Metals are assuming an increasingly pronounced role in wound healing due to the pivotal function of biofilms in chronic wound infections [28]. Metal nanoparticles are commonly harnessed to facilitate wound repair and healing, either independently or as supplementary therapeutic carriers. The most extensively investigated variants include gold [29], silver [30], copper [31], iron [32], and zinc [33], among others. Given the ongoing clinical use of numerous metal nanoparticles and the application of IONPs in regenerative medicine, coupled with the exceptional biocompatibility of IONPs in both in vivo and in vitro settings, the combination of antibacterial and biocompatible properties positions IONPs within the realm of wound healing—a facet that certain highly cytotoxic nanoparticles (such as zinc oxide nanoparticles) lack [34,35,36,37]. This paper’s objective is to investigate the role of IONPs in chronic wound healing, particularly pertaining to IONPs themselves, the correlation between IONPs and wound healing, and the underlying mechanism of their healing action.

**Figure 2 pharmaceutics-15-02327-f002:**
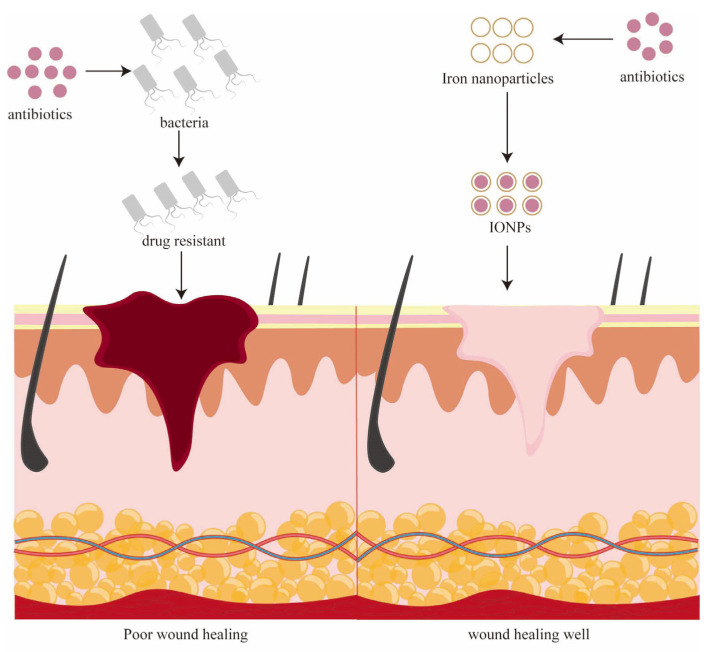
Antibiotics are still used as the primary defense against bacterial infections. However, antibiotic misuse has led to the emergence of drug-resistant bacteria, potentially spiraling localized infections out of control within chronic wounds. Metallic nanoparticles, such as iron oxide nanoparticles, offer a countermeasure to antibiotic overuse, restraining bacterial infections and fostering wound healing.

**Figure 3 pharmaceutics-15-02327-f003:**
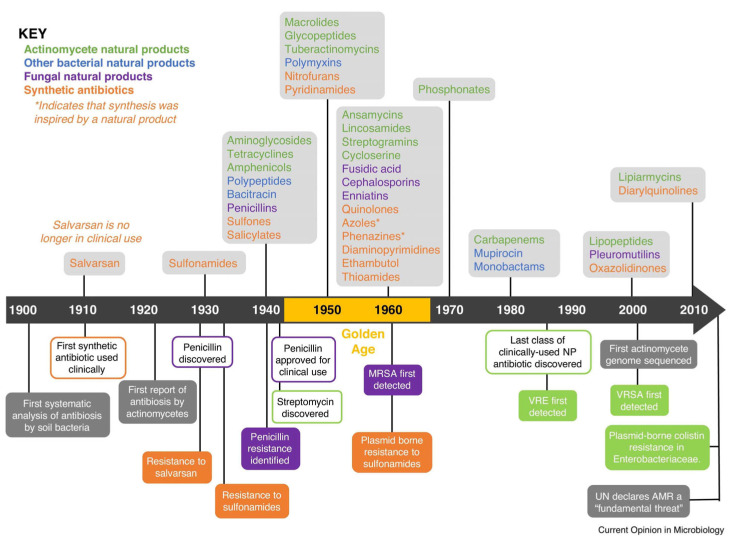
Timeline of new antibiotics entering the clinic. Antibiotics are colored according to their source: green = actinomycetes, blue = other bacteria, purple = fungi, and orange = synthetic. Essential dates pertaining to antibiotic discovery and antimicrobial resistance are provided at the timeline’s base. * Indicates that synthesis was inspired by a natural product. Source: Adapted with permission from earlier studies [38], Elsevier.

## 2. Iron Nanoparticles

### 2.1. Overview of Iron

Iron pervades nature, constituting 4.75% of crustal content—the fourth most abundant element following aluminum, oxygen, and silicon. Iron is constituted by iron atoms, forming a metallic crystal composed of metal cations and free electrons. While pure iron appears silvery white, it is often referred to as a “ferrous metal” due to the black film of tri-iron tetroxide that often covers its surface. With an atomic number of 26, a chemical formula of Fe, and an average relative atomic mass of 56, iron is a metallic element. Pure iron possesses a metallic luster and a melting point of 1538 °C, with a boiling point of 2750 °C. It is soluble in strong and medium-strong acids but insoluble in water. Iron holds valences of 0, +2, +3, +4, +5, and +6, with +2 and +3 being more common and +4, +5, and +6 being less prevalent. Notably, iron is present in the human body, and the +2-valent ferrous ion is integral to hemoglobin, facilitating oxygen transport.

Iron oxide within nanomaterials represents a widespread natural compound, including 16 iron oxide-like entities, including oxides, hydroxides, and oxide-hydroxides. Their basic components involve Fe, O, and/or OH, yet variations in chemical valence and crystal structure occur within Fe (Table 1). Key iron oxides include magnetite (Fe_3_O_4_), magnetic hematite (γ-Fe_2_O_3_), and hematite (α-Fe_2_O_3_), notably Fe_3_O_4_ and γ-Fe_2_O_3_, extensively used in biomedical applications.

IONPs mainly consist of nanomaterials composed of magnetic hematite (γ-Fe_2_O_3_) and/or magnetite (Fe_3_O_4_) particles, displaying hexagonal structures with diameters ranging from 1 to 100 nm [39]. IONPs find application as catalysts [40], pigments [41], sensors [42], heavy metal removal agents [43], antibacterial agents [44], and cancer treatment [45], among other roles. Due to their smaller particle diameter, greater surface area-to-mass ratio, and enhanced activity compared to traditional metal ions, IONPs exhibit exceptional antibacterial performance. They adhere more effectively to bacterial surfaces, allowing greater penetration of metal ions through cell walls and membranes, thereby exhibiting potent bactericidal effects [46,47].

Magnetic nanoparticles typically under 30 nm in diameter display superparamagnetism, with superparamagnetic iron oxide nanoparticles (SPIONs) exemplifying nanoscale particles with magnetic responsiveness. When magnetic nanoparticle sizes dip below the superparamagnetic threshold, they acquire supermagnetic properties. Magnetite (Fe_3_O_4_), magnetic hematite (γ-Fe_2_O_3_), and hematite (α-Fe_2_O_3_) serve as chief iron oxides, not only manifesting superparamagnetism but also demonstrating commendable biocompatibility [48]. SPIONs notably possess the superparamagnetic property of generating heat under magnetic field influence and can be guided to specific tissues using magnetic fields. SPIONs find widespread application in imaging [49], tissue repair and cell differentiation [50], immunoassays [51], thermal therapy, and drug delivery [52,53], among others (Table 2).

Consequently, it is evident that iron nanoparticles are predominantly represented by IONPs. Within this category, SPIONs offer specific attributes, rendering them promising candidates for diverse research avenues (Table 2).

### 2.2. Iron Is Involved in Wound Reconstruction

Iron plays a crucial role in the body’s wound-healing process. It holds significance in maintaining human health, constituting 0.01% of the total body weight. Notably, iron is predominantly present within the skin, the body’s largest organ, given that approximately one-fifth to one-quarter of absorbed iron is eliminated daily due to shedding epidermal cells [54]. In order to mitigate iron’s potential toxicity, the body converts it into ferritin and iron-containing hemoglobin, effectively segregating it from normal cellular components.

Clinical observations underscore a strong connection between chronic wounds and iron. For instance, ulcers exemplify this relationship, as the healing of chronic wounds deviates from normalcy when the body’s iron levels are abnormal. Insufficient iron levels affect hemoglobin synthesis, leading to the emergence of iron deficiency anemia in patients. In the case of diabetic patients, the individuals themselves encounter risks of progressive lesions and compromised functionality in blood vessels, nerves, kidneys, and other tissues and organs (Figure 4). Notably, clinical research has established a correlation between the severity of diabetic foot ulcers and declining hemoglobin levels [32]. Pertinent studies affirm a direct and significant correlation between anemia and wound healing; patients with anemia generally exhibit subpar wound healing compared to those without anemia [55]. Predominantly, anemia can be attributed to iron deficiency and diminished iron reserves in the body, alongside other iron-related factors [56]. It is noteworthy that iron deficiency anemia is more prevalent in diabetic patients in comparison to non-diabetic individuals [57,58,59,60]. When iron deficiency anemia coexists with a diabetic ulcer (diabetic foot), peripheral vasculopathy’s severity exacerbates, potentially even escalating the risk of necrosis and limb amputation among diabetic patients. Concurrently, compromised vascularity and diseased nerves can perpetuate iron deficiency anemia, thereby engendering a reciprocal relationship that sustains the diabetic ulcer [61]. In instances of iron overload within the body, excessive iron deposition in tissues aligns with pronounced skin damage. Excessive iron accumulation in local skin and macrophages can impede wound healing. Iron functions as a mediator of skin toxicity across various pathological conditions, including inflammation, infection, cancer, and sunburn. In chronic venous disease (CVD), excessive iron accumulation accompanies increased skin damage. Notably, iron-containing heme serves as a histological indicator of tissue iron excess. Within the context of CVD, iron ions within iron-containing heme contribute to the disease’s pathogenesis. This occurs through pathways involving free radicals, metalloproteinases, etc., ultimately leading to stromal degradation and ulcer development driven by iron ions [62].

The mechanisms orchestrating the process of wound healing include several key components. These include vasoconstriction, the contraction of the wound itself, the involvement of inflammatory mediators and chemokines, as well as interactions between cells and the extracellular matrix (Figure 5). Furthermore, the proliferation and modification of various elements, such as skin, blood vessels, and tissues, play significant roles in this intricate process [18]. Iron plays a role in the body’s oxygen and energy metabolism, synthesizing hemoglobin and myoglobin for oxygen transport and storage, participating in the energy metabolism of highly active organs, and engaging in cellular metabolism and other bodily metabolic processes. 

**Figure 5 pharmaceutics-15-02327-f005:**
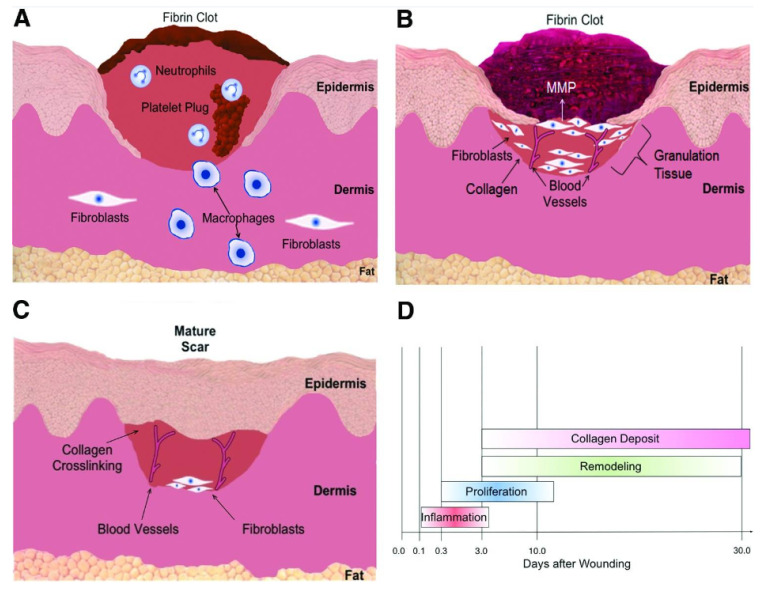
(**A**) Inflammation. Hemostasis and inflammation can occur immediately after an injury. Blood extravasation leads to the formation of blood clots. Many signaling factors are released, causing inflammatory cells like neutrophils and monocytes to be re-attracted to the wound. The monocytes then differentiate into mature macrophages at the wound site. Both neutrophils and macrophages are phagocytic cells that act to cleanse the wound while releasing other factors that stimulate fibroblasts to converge at the wound site. (**B**) Proliferation and remodeling. During proliferation, fibroblasts secrete the extracellular matrix and form granulation tissue. Angiogenesis unfolds concurrently with the migration of endothelial cells to the wound’s vicinity. As an integral facet of the remodeling phase, matrix metalloproteinases concomitantly degrade the collagen secreted by fibroblasts. (**C**) Maturation ensues, wherein collagen synthesis and degradation achieve equilibrium. Disorderly collagen fibers undergo crosslinking and alignment along tension lines, consequently amplifying the wound’s tensile resilience. (**D**) Chronological progression of diverse processes in wound healing. Source: Adapted with permission from earlier studies [63], Advances in Wound Care.

Iron deficiency may affect wound healing through various mechanisms, and it can influence hemoglobin synthesis, leading to iron deficiency anemia as well as hypoxia, reduced immunity, and several other adverse consequences. Among these, hypoxia plays a key role in wound healing. Hypoxia-inducible factor-1 (HIF-1) is of great importance at any stage of wound healing due to its role in cell migration, cell survival under hypoxic conditions, cell division, growth factor release, and matrix synthesis (Figure 6). Positive regulators of HIF-1, such as prolyl-4-hydroxylase inhibitors, have shown benefits in enhancing ischemic wound closure in diabetes. Conversely, HIF-1 deficiency can lead to chronic hypoxia, which contributes to the formation of hard-to-heal ulcers [63]. Furthermore, good wound healing necessitates an ample supply of energy, of which a sufficient and efficient blood supply is essential—providing adequate oxygen and various nutrients for wound healing. Anemia in diabetic patients might be related to the fact that inflammatory cells in the patient’s body hinder the transport of iron to the bone marrow. The available iron gets aggregated by macrophages and stored in ferritin. While the aggregate iron content stored within the body remains unaltered, the associated iron becomes unavailable for utilization in physiological functions. This circumstance can induce functional iron deficiency, culminating in the manifestation of iron deficiency anemia. Reduced blood flow, decreased oxygenation due to anemia, and inadequate local nutrition can delay or even halt wound healing. Similarly, iron overload can have adverse effects on the body. For instance, in cases of chronic venous disease (CVD), the iron content of the skin increases. Excess iron in tissues is stored in the form of iron-containing hematoxylin, which serves as a histological marker of excessive iron in tissues. According to current experimental studies, significant deposits of iron-containing heme appear in skin lesions and ulcerated tissues, while normal skin and regenerated dermal tissue in wounds are largely devoid of iron-containing heme [64]. In CVD, iron is released due to excessive oxidative stress, generating large amounts of free radicals. These radicals activate the hydrolysis of metalloproteinases, making them overactive, and downregulate the levels of tissue inhibitors of metalloproteinases. Additionally, the local tissues of patients with ulcers might have lost the ability to counteract the internal iron output of macrophages, possibly due to genetic factors [65].

**Figure 6 pharmaceutics-15-02327-f006:**
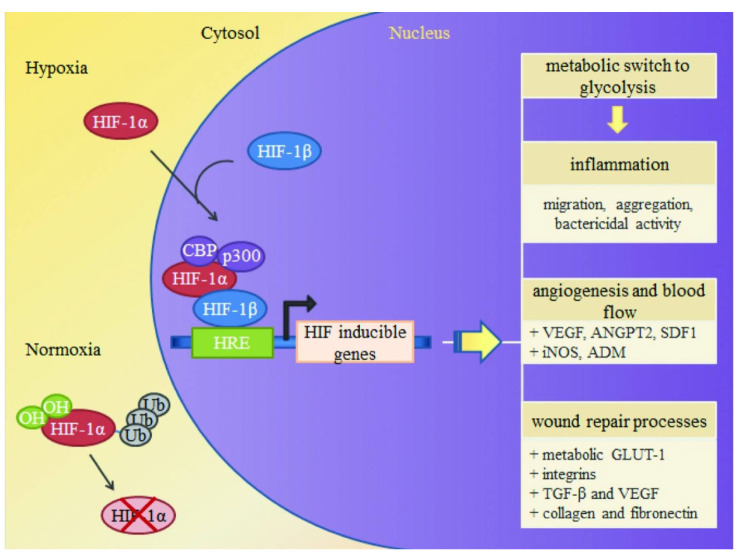
Activation of HIF-1. Under hypoxia, HIF-1α stabilizes and binds to the HIF-1β subunit, forming the active transcription factor HIF-1. HIF-1 translocates to the nucleus, where it binds to hypoxia regulatory elements within the promoter region of HIF-1-inducible genes, along with coactivators p300 and CBP, to promote the expression of HIF-1 target genes, such as VEGF and SDF-1. Source: Adapted with permission from earlier studies [63], Advances in Wound Care.

Regarding the relationship between iron and wounds, it is known that abnormal iron levels (whether low or high) can lead to poor wound healing. Abnormal iron levels are often linked to factors such as local tissue nutrition and blood flow, which directly or indirectly hinder the normal wound healing process.

## 3. Iron Nanoparticles for Chronic Wounds

The ramifications of metallic elements like iron and copper on human and animal biology have been extensively deliberated and investigated by concerned individuals [66,67]. The antimicrobial efficacy of metal nanoparticles represents a multifaceted approach against bacteria, thereby mitigating the emergence of drug resistance. These nanoparticles exhibit the capacity to combat multidrug-resistant bacteria by impeding respiration and hindering cellular growth. In the realm of clinical medicine, iron oxide nanoparticles find utility across diverse applications, including imaging, oncology, immunology, and regenerative medicine, among others [68,69,70,71,72,73,74,75,76,77].

The application of IONPs in chronic wound healing primarily revolves around their utilization as wound dressings in surgical contexts. Addressing chronic wounds mandates tailoring treatments according to their etiology while concurrently addressing underlying systemic and local factors. For instance, diabetic patients necessitate diligent blood sugar management and real-time kidney function monitoring. In this regard, trauma management emerges as a significant therapeutic tool. The optimal surgical dressing should exhibit robust antimicrobial and anti-infection properties, facilitate wound healing and repair, ensure cost-efficiency, and minimize harm to both the wound and the organism. Traditional wound dressings, such as gauze, fulfill fundamental wound healing requisites, but their susceptibility to secondary damage during dressing changes can detrimentally affect subsequent wound healing and exacerbate existing damage. By incorporating IONPs as reinforcing materials within dressings, the dressing’s antibacterial and infection control capabilities witness substantial enhancement due to the broad-spectrum antibacterial attributes of nanoparticles [50]. Furthermore, their physical and chemical attributes contribute to diminishing antibiotic-resistant bacterial strains, consequently enabling antibiotic dosage reduction (Figure 7) [78]. IONPs have the potential to serve as dressings for chronic wounds in diabetic patients. Diabetes mellitus engenders chronic wounds that pose intricate challenges in terms of healing, stemming from hyperglycemia, malnutrition, anemia, and hypoxia. Beyond their intrinsic antimicrobial properties, IONPs possess the capacity to rectify local tissue anomalies, including nutritional deficits, ischemia, and hypoxia, through iron release [79]. Moreover, IONPs demonstrate positive effects on diabetic patients. For instance, these nanoparticles exhibit inhibitory effects on alpha-amylase, thereby assisting in regulating the blood sugar levels of diabetic patients (Figure 7) [2,80]. Notably, superparamagnetic iron oxide nanoparticles (SPIONs) have been found to significantly lower blood glucose levels in diabetic rats, surpassing the effects of metformin in initial animal models of diabetes research [81].

The research conducted by Mario Ledda et al. has demonstrated the absence of acute or chronic toxicity in mice for up to seven weeks following intravenous administration of SPIONs [82]. Combining the use of Fe_2_O_3_Nps in regenerative medicine with IONPs’ commendable in vitro and in vivo biocompatibility, which is notably absent in highly cytotoxic zinc oxide nanoparticles [34,35,36,37], underscores the synergistic potential of antimicrobial properties and biocompatibility that positions IONPs as valuable entities in wound healing.

### 3.1. Mechanism of Iron Oxide Nanoparticles in Wound Healing

#### 3.1.1. Antibacterial Activity

The primary avenue through which IONPs contribute to wound healing lies in their antibacterial efficacy. A causal link exists between wound infection and chronic wounds, where wound infection ranks among the most prevalent and significant complications of chronic wound healing, often obstructing the progress of wound healing. Earlier studies have unveiled the inhibitory effect of IONPs and iron NPs on bacterial growth (Figure 8) [83,84]. Guo J. et al. established a wound infection model in mice involving Staphylococcus aureus. Subsequently, the experimental group demonstrated significantly fewer residual bacteria than the control group, thereby affirming the in vivo antibacterial impact of IONPs [85]. Simultaneously, chitosan-coated Fe3O4NPs exhibit commendable sterilization capabilities [86]. When deployed as dressing materials, they elicit robust antibacterial and antifungal effects.

The antibacterial attributes of IONPs hinge not only on oxide morphology but also on nanoparticle size, morphology, and other physicochemical attributes [2,47]. Upon entering cells, IONPs can release a substantial volume of free radicals (ROS) [87]. Nanoparticles, due to their size, are susceptible to engulfment by phagocytic cells residing alongside target microbes. Once within the cell, the nanoparticle can dispense drugs to incapacitate the microbe. Furthermore, smaller particle sizes and larger surface area-to-mass ratios endow IONPs with an effective biofilm prevention capacity.

**Figure 8 pharmaceutics-15-02327-f008:**
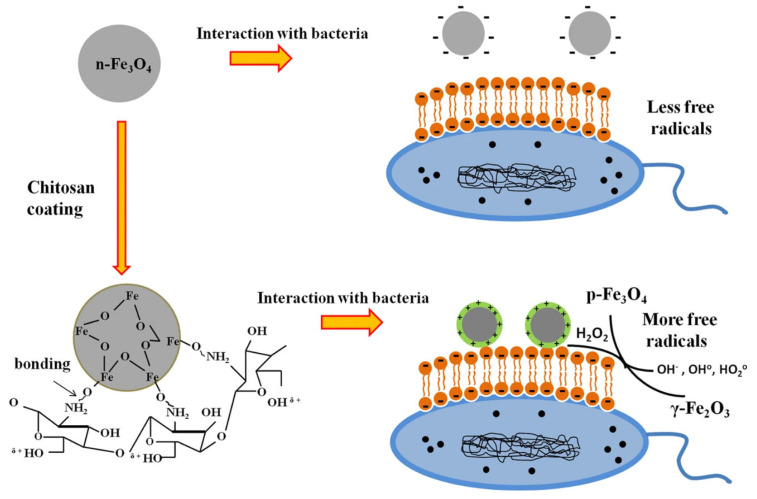
Comprehensive mechanism of IONPs against bacterial cells. Source: Adapted with permission from earlier studies [83], Springer Nature.

Following cell surface contact, IONPs initiate interactions with fundamental bacterial cell constituents such as DNA, lysosomes, ribosomes, and enzymes. This engagement leads to oxidative stress, heterogeneity alterations, shifts in cell membrane permeability, electrolyte balance, enzyme inhibition, protein inactivation, and gene expression modifications.

The initial facet involves the interaction of NPs with the cellular barrier. The cell wall and cell membrane assume pivotal roles as barriers for bacteria, concurrently influencing significant physiological activities of bacteria. IONPs particles, characterized by a diminutive diameter, volume, and extensive surface area-to-mass ratio, enable enhanced adherence to the bacterial surface [46,47]. By virtue of their nanoscale dimensions, iron oxide nanoparticles have the capability to permeate bacterial cell walls after adhering to cell membranes. Consequently, they possess the capacity to modify the membrane structure, potentially leading to degeneration of the plasma membrane, culminating in organelle rupture and even cell lysis. Bacterial growth and reproduction hinge upon signal transduction, wherein iron oxide nanoparticles disrupt bacterial signaling through their influence on phosphorylation and protein denaturation during bacterial signaling. This disruption results in apoptosis and the eventual cessation of bacterial proliferation. IONPs heighten the permeability of the cellular barrier through electrostatic attraction and an affinity for thionin. This fosters superior adhesion of metal ions to bacterial cell walls and plasma membrane surfaces, thereby compromising the integrity of bacterial cell membranes [88,89]. IONPs exhibit antibacterial effects on both Gram-positive and Gram-negative bacteria [44,90,91,92]. Some studies underscore the more pronounced antibacterial effects of IONPs on Gram-negative bacteria [93], largely attributed to the composition of the cell wall in Gram-negative bacteria—comprising LPS, lipoproteins, and phospholipids—which collectively form a penetration barrier. In stark contrast, the cell wall of Gram-positive bacteria features a thin layer of peptidoglycan and phosphocholic acid, alongside abundant stomata. Gram-negative bacterial cell walls include lipopolysaccharides, predominantly consisting of lipids and polysaccharides, which offer a negatively charged region that attracts IONPs. Within the realm of Gram-positive bacteria, teichoic acid holds a distinct position as a cell wall component. This polymer consists of ribitol and/or glycerol residues interconnected through phosphodiester bonds. Moreover, in contrast to Gram-negative bacteria, Gram-positive bacteria exhibit increased negative charges on their cell wall surfaces, further enhancing their attraction to NPs [94].

The second facet includes oxidative stress. Iron, as a transition metal, participates in both electron donation and acceptance. Consequently, excessive iron content can prove cytotoxic. In natural and biological environments, ferrous iron (Fe^2+^) interacts with hydrogen peroxide (H_2_O_2_), producing OH• free radicals. Intracellular excesses of iron are sequestered by ferritin, yet free iron retains the potential to exert toxic effects and participate in the Fenton reaction, thereby engendering reactive oxygen species (ROS) production. ROS-induced oxidative stress stands as a pivotal antimicrobial mechanism of metal nanoparticles. Iron has been documented to generate ROS and disrupt electron transport, culminating in superoxide production [95]. Furthermore, IONPs can generate hydroxyl radicals (-OH), inducing acute microbial death. Under normal conditions, bacterial cell ROS production and scavenging remain in equilibrium. However, excessive ROS production disrupts cellular redox balance, resulting in oxidative stress that impairs individual bacterial cell components [96,97]. ROS exacerbates mitochondrial outer membrane permeability, damages lysosomal membranes, and triggers iron release from these organelles. The ensuing ROS interaction with the cell membrane precipitates lipid peroxidation, where the resultant toxic products interact with and incapacitate proteins. Compromised proteins evade recognition by the ubiquitin/proteasome system. ROS instigate cell damage by modifying lipids, proteins, and DNA, or they may trigger secondary ROS production, culminating in cell demise.

IONPs possess intrinsic physical attributes that confer remarkable antimicrobial prowess, thereby facilitating the management of bacterial infections within wounds. Furthermore, IONPs are capable of curbing bacterial infection by means of their interaction with cellular barriers and the imposition of oxidative stress. Resistance mechanisms chiefly include the deactivation of antimicrobial agents due to the production of enzymes by bacteria, rendering these agents inert. Alteration of bacterial outer membrane permeability also contributes to resistance. In the context of iron oxide nanoparticles, their ability to heighten the permeability of cell membranes serves as a countermeasure against bacterial resistance during extended usage. When employed as drug carriers, IONPs efficaciously thwart the enzymatic degradation of antibiotics prior to their bactericidal action, thereby impeding the emergence of acquired resistance. Combining IONPs with antibiotics as carriers optimizes the management of resistance, penetration, and biofilm eradication. Resistance to long-term utilization of metal nanoparticles primarily stems from biofilm development. IONPs exhibit increased biofilm penetration capabilities, especially when an external magnetic field is introduced [98]. This non-antibiotic pathway of bacterial inhibition, demonstrated by metal nanoparticles, serves as an efficacious strategy to combat antibiotic abuse and enhance wound dressings in the context of contemporary antibiotic usage practices.

#### 3.1.2. Release of Metal Ions

Metal ions are capable of gradual release from metal oxides, infiltrating cells via the cell membrane. This interaction influences enzyme activity, disrupts normal cell structure, and impedes regular cellular physiological processes. Wounds that endure prolonged healing periods often stem from aberrations such as bacterial infection, local tissue ischemia, and hypoxia. In the case of chronic wounds, including diabetic wounds, systemic factors such as anemia and hyperglycemia are also significant contributors to hindering wound healing. IONPs facilitate the release of iron, which can exert a modulating effect on anemia and abnormal blood sugar in certain patients, such as those with diabetes. Notably, Anbazhagan Sathiyaseelan’s in vitro wound healing scratch experiment using CS/PV A-PD-FeO NPs (0.01%) has verified the role of IONPs in diabetic wound healing [2]. By releasing iron, IONPs increase local wound and surrounding tissue replenishment. Their inherent properties enable them to enhance cell membrane permeability, allowing them to traverse barriers and reach tissues and blood vessel interiors, thus rectifying localized iron deficiency (Table 3) [94]. This gradual and controlled iron release, facilitated by the structural constraints of IONPs, effectively circumvents iron overload and potential biotoxicity (Table 3) [2]. Additionally, iron nanoparticles exhibit dose-dependent α-amylase inhibitory activity, a factor contributing to improved blood glucose levels in diabetic patients. Notably, SPIONs have demonstrated superiority over metformin in certain aspects through animal model experiments. In vitro, release profiles indicate a low percentage of sustained iron release over 24 h. This restrained iron release characteristic of the nanocomposite offers the advantage of avoiding iron overload-associated toxicity, rendering it beneficial for the process of diabetic wound healing related to anemia (Table 3).

Moreover, IONPs manifest a certain inhibitory effect on α-amylase during in vivo cellular physiological processes, thereby aiding in the regulation of blood sugar levels in diabetic patients [2,80].

#### 3.1.3. Overcoming Antibiotic Resistance

Antibiotic resistance poses a grave concern. This phenomenon involves pathogenic microorganisms, including bacteria, parasites, and viruses, evolving to resist antibiotics to adapt to their environment and counteract drug effects. Pervasive antibiotic use and the emergence of antibiotic-resistant pathogens have engendered a global health issue necessitating urgent alleviation and resolution. The integration of IONPs into chronic wound treatment could be a pivotal solution to this quandary.

In the context of applying iron nanoparticle dressings to chronic wounds, their potent antibacterial activity complements antibiotics. Capitalizing on the physical and chemical attributes of IONPs, such as their diminutive size, volumetric extent, extensive surface area, and inherent vigor, they serve not only as autonomous antibacterial agents but also as carriers and mediators of antibiotics (Figure 9). Consequently, coupling IONPs with wound dressings and antibiotics enables dosage reduction while maintaining effective antibacterial potential, thereby mitigating pathogen resistance issues within wounds [99]. Extant research indicates that nanoparticle-anchored antibiotics enhance local antibiotic concentration and antibacterial efficacy. Simultaneously, IONPs facilitate antibiotic penetration through cell membranes and walls, optimizing bactericidal objectives.

In the case of commonly encountered drug-resistant bacteria, like Staphylococcus aureus, IONPs exhibit discernible direct bactericidal influence [28,90]. Furthermore, IONPs have the capacity to impede Staphylococcus aureus biofilm formation [46].

Among metal nanoparticles, IONPs presently stand as a suitable choice for wound dressing, owing to their robust anti-infective attributes and their potential as drug delivery carriers. IONPs possess distinctive advantages over alternative metal nanoparticles. While numerous studies on metal nanoparticles center on combating bacterial infections, they frequently overlook the comprehensive and dynamic nature of wound healing, which often encounters interference from factors like poor local oxygenation and systemic conditions such as diabetes and anemia. Through iron release, IONPs not only regulate local blood oxygen levels but also rectify specific populations (diabetic) anemia and blood glucose aberrations. The controlled release of iron from nanocomposites circumvents iron overload toxicity, rendering it particularly advantageous for the healing trajectory of anemia-associated diabetic wounds. Unlike some other metal nanoparticles, IONPs exhibit moderate toxicity towards eukaryotic cells and certain bacteria [100]. Although diverse metal nanoparticles prove efficacious in combating bacterial infections, the attributes of IONPs, including their lower toxicity to eukaryotic cells and relative harmlessness to certain bacteria, position them as a prime choice for holistic wound healing from a perspective including toxicity, biocompatibility, and overall bodily harmony [101]. This assumes paramount significance, especially amid the prevailing backdrop of antibiotic misuse, where IONPs’ advantages become a potent strategy for propelling wound healing. 

### 3.2. Mechanism of Superparamagnetic IONPs in Wound Healing

Amid diverse metal oxide nanoparticles, IONPs have garnered prominence due to their superparamagnetic attributes, which are subject to alignment by external magnetic fields. Among these, Fe_3_O_4_ NPs exhibiting increased paramagnetic characteristics are labeled as superparamagnetic IONPs (SPIONs). In addition to the aforementioned antibacterial features, IONPs—owing to the distinctive ordered structure of FeO•Fe_2_O_3_ intrinsic to Fe_3_O_4_—can evince magnetic properties when subjected to a magnetic field.

#### 3.2.1. Magnetic

Fe_3_O_4_ NPs have drawn substantial interest within the realm of magnetic fields, primarily due to their superparamagnetic characteristics and high magnetic susceptibility [52,102,103]. In comparison to their nickel- and cobalt-based counterparts, IONPs exhibit reduced toxicity, thus positioning them as the favored choice among magnetic nanoparticles [104]. Given the magnetic attributes associated with IONPs-based magnetic nanoparticles, SPIONs, they manifest intrinsic responses under high-frequency alternating magnetic fields. This leads to outcomes such as shock damage, local hyperthermia, and ROS generation, culminating in biofilm disruption and cell demise [99,105]. Moreover, SPIONs can directly traverse cell membranes and engage with microbial cells under the influence of an external magnetic field. The manipulation of interactions between SPIONs can be achieved through externally applied magnetic field gradients. These gradients facilitate their concentration, along with their carriers, at designated sites. Employing diverse magnetic fields generates field gradients that act upon nanoparticles, enabling their movement along the force imparted by the magnetic field. Combinations of various field gradients permit IONPs to alter their position and trajectory. The combination of field gradients, as elucidated by the magnetophoretic model, fundamentally determines the resultant directional force experienced by the MNPs and their ensuing motion [106]. Advanced supermagnets enable the creation of finer magnetic field gradients within the body’s trauma sites. This enhanced precision facilitates more targeted treatment, thereby markedly enhancing patient management and healthcare resource allocation optimization. External coatings can also be applied to nanoparticles to enhance stability and provide a platform for IONPs’ functionalization [107,108].

#### 3.2.2. Functionalized SPIONs Accelerate Wound Healing

IONPs can be incorporated into diverse biomaterials to craft functionalized external dressings. Capitalizing on their intrinsic responses to external magnetic fields, these dressings can foster the generation of specific bioactive substances conducive to wound healing. Through functionalization, SPIONs can serve as antibiotic carriers, releasing antibiotics into targeted subcellular organelles of bacteria under the influence of an external magnetic field (Figure 9). This process can inflict damage on organelles and even lead to cell lysis [109]. Alternatively, drugs can be delivered to areas of inflammation or bacterial cells, effectively eradicating bacteria and pathogens through localized high drug concentrations and curbing the emergence of drug resistance. Notably, Jiang Wu et al. fabricated functionalized IONPs loaded with fibroblast growth factor (b-FGF) onto Fe_3_O_4_ NPs. This combination enabled sustained b-FGF release, guided by a specific-frequency external magnetic field (e-MF) [110]. In vitro studies underscored that such functionalized iron nanoparticles could transform macrophages from a pro-inflammatory phenotype (M1) to an anti-inflammatory, healing-promoting phenotype (M2) under external magnetic field guidance [111]. Additionally, animal models featuring full-thickness wounds were employed to assess wound healing effects. Findings demonstrated that the developed functionalized IONPs synergistically improved wound healing through M2 macrophage polarization and sustained b-FGF release, consequently expediting wound healing.

Numerous studies indicate that functionalized IONPs, in conjunction with other biomaterials, can exhibit numerous advantageous traits. For instance, these nanoparticles can improve wound dressing fibers’ resistance to fungal cell adhesion and biofilm formation [46]. Thus, functionalized IONPs emerge as a promising avenue for wound dressings, medical devices, and other clinically relevant materials.

#### 3.2.3. Overcoming Antibiotic Resistance

Additionally, IONPs, when subjected to an external magnetic field, can enhance antibiotic delivery to target sites by virtue of their magnetic properties. This aids in preserving drug integrity, maximizing efficacy, and reinforcing bactericidal potency. Guided by an external magnetic field or specific stimuli, IONPs can release antibacterial drugs at relatively high concentrations precisely at the locations of target pathogenic microorganisms or biofilms. This is achieved while maintaining a low total dose, which minimizes the likelihood of antibiotic resistance and reduces adverse effects on the human body. Simultaneously, high localized concentrations effectively neutralize target microorganisms [112,113].

Research reveals that among various metal nanoparticles, SPIONs featuring distinct surface coatings (e.g., gold or silver) exhibit the highest antibacterial activity against biofilms [112]. Moreover, under the influence of an external magnetic field, magnetic metal nanoparticles possess the remarkable capability to penetrate biological membranes [99,105].

IONPs execute bacterial eradication through multiple mechanisms while reinforcing the barrier against bacterial resistance. This renders metal nanoparticles more resilient to bacterial adaptation compared to antibiotics, which typically target microorganisms through a single pathway [114]. Additionally, SPIONs can be functionalized and directed by an external magnetic field for enhanced drug delivery precision and synergistic effects on local cells.

## 4. Application of Iron Nanoparticles in Wound Healing in a Clinical Setting

Nanoparticle dressings have been employed to alleviate patient discomfort, manage infections, and expedite wound recovery due to their cost-effectiveness, antibacterial properties, stability, and bioactivity [43]. As chronic wounds gain attention, effective strategies for their management are being sought. For chronic wound treatment, timely and efficient debridement is crucial for optimal wound closure and faster healing [115]. IONPs have captured the attention of scientists due to their superparamagnetic characteristics and biomedical potential stemming from their biocompatibility and non-toxic nature. The primary objective of treating chronic wounds is to combat infections. Antibacterial drugs are commonly used in clinical practice. Ferromagnetic nanoparticles, such as Fe_3_O_4_, have been employed to combat bacterial biofilms. These nanoparticles utilize peroxidase-like activity to eliminate bacteria, break down existing biofilms, and prevent biofilm formation [116]. IONPs also play a significant role in regenerative medicine. In simulated conditions using a rat trauma healing model, bone marrow MSC-derived exosomes treated with IONPs and static magnetic fields have been shown to accelerate trauma closure, reduce scar width, and enhance angiogenesis, all of which have been demonstrated in clinical settings [117].

With growing concern about infections, researchers are exploring novel approaches to addressing wound infections, including the role of wound dressings. Wound dressings are applied to wounds to remove excess fluids, protect against infection, and aid in wound healing. Proper use of wound dressings can accelerate healing and reduce complications. In the context of infection management, antibiotics are used on wound dressings to target wound pathogens. However, concerns about antibiotic resistance have led to a focus on non-antibiotic strategies.

In current wound therapy, the application of specialized biocompatible nanoparticles that carry and release bioactive drugs in a controlled manner has been shown to significantly improve wound healing. Additionally, the intrinsic properties of metal nanoparticles can induce biological responses such as antibacterial activity and antioxidant properties. Researchers have developed wound dressings based on filamentous protein-IONp scaffolds that are compatible with human adipose-derived stem cells. Other studies have explored nanocomposite scaffolds composed of chitosan (CS), polyvinyl alcohol (PVA), and iron oxide nanoparticles (FeO NPs) for drug delivery with antioxidant, antibacterial, and antidiabetic effects [2]. These concepts align with the use of porous structures in clinical wound dressings that maintain a moist environment, promote breathability, and absorb fluids during healing. Numerous clinical studies and practices have unequivocally demonstrated that attaining a high degree of biocompatibility and mitigating cytotoxicity are the guiding principles for contemporary wound dressing advancements.

## 5. Limitation

### 5.1. The Toxicity of Some Eukaryotic Cells Is Relatively Weak to Some Bacteria

The constrained toxicity of IONPs towards particular eukaryotic cells is observed in contrast to certain bacterial strains. On the one hand, the limitation of IONPs is discernible in their relatively feeble antibacterial efficacy against certain bacteria when compared with other metal nanoparticles. For instance, Fe_2_O_3_ NPs exhibit lesser potency than ZnO and CuO NPs against Escherichia coli, Staphylococcus aureus, Pseudomonas aeruginosa, and Bacillus subtilis. This variance might stem from the distinct antimicrobial capabilities inherent in diverse metals. Another potential explanation lies in the indispensability of Fe^2+^ for bacterial proliferation. While an excess of Fe^2+^ can hinder bacterial physiological activities, the influence of IONPs on bacterial proliferation cannot be unequivocally dismissed. Relevant investigations indicate that exposure to sub-inhibitory concentrations of Moxicillin combined with IONPs in the presence of humic acid increases the bacterial growth of Pseudomonas aeruginosa and Staphylococcus aureus [118].

### 5.2. Toxic Effects on Some Eukaryotic Cells

On the flip side, IONPs exhibit toxicity toward specific eukaryotic cells [119]. Some researchers posit a correlation between toxicity and the size of iron oxide nanoparticles, even if the relationship is not linear. Generally, diminutive IONPs, such as superparamagnetic iron oxides with a particle size less than 2 nm (SPIONs), possess a greater propensity for permeating cellular membranes and subsequently disrupting intracellular organelles, thereby potentially inducing toxicity [120]. The chief mechanism underlying IONP toxicity involves the generation of reactive oxygen species (ROS), culminating in escalated lipid peroxidation that compromises cellular membrane function and intracellular components, including DNA, organelles, and proteins, and ultimately elicits inhibitory consequences [121]. Notably, iron accumulation at the target site contributes to SPION-associated toxicity, precipitating iron overload—a condition characterized by excessive iron deposition within the body, triggering structural impairment and malfunction of vital organs such as the heart, liver, and pituitary gland. Clinically, this manifests as heart failure, hepatic fibrosis, developmental anomalies, and potentially fatality [122].

Consequently, judiciously managing biological toxicity constitutes an imperative concern for SPIONs. Within the theoretical framework of clinical utilization, toxicity control is delineated into three stages: ① Material Fabrication: Exercising control over particle dimensions stands as the foremost and standardized approach for SPIONs [120]. An optimal nanoparticle size not only circumvents potential toxic repercussions but also enhances clearance mechanisms, including macrophage-mediated phagocytosis and renal excretion [48]. Gradual iron release from nanocomposites not only mitigates ROS-mediated damage but also prevents iron overload at the site of action. ② Patient Selection: Prior to deploying iron oxide nanoparticles and their derivatives for wound healing, a proactive and meticulous assessment of patients’ medical histories is essential. This approach serves to exclude individuals with recent extensive blood transfusions and tailor personalized transfusion protocols for patients necessitating blood transfusions. ③ Prognostic Monitoring and Intervention: Regular serum ferritin (SF) assessment is advisable, with the option of employing desferrioxides as required [123].

### 5.3. Limitations in the Current State of Research on Iron Nanoparticles for Wound Healing

In ongoing investigations, limitations primarily pertain to material preparation and external magnetic fields. Particularly, a critical reevaluation of SPION preparation methods is warranted. Numerous scholars are dedicated to refining SPION synthesis methodologies; however, the characterization and standardization of SPIONs necessitate further exploration and practical implementation [109]. For superparamagnetic SPIONs, external magnetic fields undeniably constitute a pivotal facet of their operational paradigm. Presently, magnets and other magnetic field generators are the primary sources of such fields. Nonetheless, achieving more precise and diverse localization and trajectories of SPIONs mandates intricate magnetic field gradients, entailing substantial hardware prerequisites at the current stage of research [120]. Parallel to the ongoing enhancement of SPION synthesis techniques, considerable emphasis is being placed on devising more sophisticated and portable external magnetic field devices. The incorporation of intricate magnetic field gradients into wound dressings could potentially revolutionize the therapeutic approach to chronic wounds, ushering in a transformative era of wound management for patients.

## 6. Conclusions

IONPs harmoniously include both antibacterial prowess and biocompatibility. Dressings infused with IONPs not only facilitate wound healing by combating bacterial infections through their antimicrobial attributes but also contribute to patients’ well-being by virtue of their iron properties, subsequently promoting wound tissue restoration and restructuring through the controlled release of specific bioactive substances. Additionally, the combined application of antibiotics and iron nanoparticle-laden dressings in wound healing endeavors addresses the challenge of antibiotic resistance, enabling increased antibiotic concentrations with reduced overall consumption. This synergy improves the capacity to counter infections, yielding increased efficacy with diminished effort. Despite the aforesaid advantages of IONPs, rendering them efficacious in surgical dressings for averting and ameliorating chronic wound infections, IONPs exhibit certain limitations. On the one hand, they may not wield uniform effectiveness against all bacterial strains, and on the other hand, they could elicit deleterious effects on healthy cells. Fortuitously, these limitations can be alleviated through the incorporation of stabilizers and composites, ameliorating adverse repercussions. The multifaceted attributes of iron have spurred scholarly contemplation regarding its combination of antimicrobial and drug-delivery capacities. Generally, metal nanoparticles exhibit the potential to combat infections via non-antibiotic pathways; however, they often carry a degree of cytotoxicity, limited biological activity, and increased costs. The ideation catalyzed by IONPs ultimately circles back to the IONPs themselves: the convergence of antimicrobial action and biocompatible metal nanoparticles renders them amenable for deployment in wound dressings. Moreover, when coupled with external magnetic fields and functionalization, IONPs manifest unique release patterns, thereby offering a potent avenue for drug delivery and local wound healing promotion. Furthermore, continuous optimization of external magnetic fields presents the prospect of sophisticated, user-friendly magnetic field gradient-forming devices. Such advancements not only enhance patient compliance and ambulatory treatment but may also engender novel drug delivery regimens.

Hence, we believe that IONPs play an indispensable affirmative role in chronic wound healing. Through synergistic combinations with diverse biomaterials, IONPs can yield functionalized derivatives tailored for personalized and targeted treatment of distinct wound conditions. This paradigm extends beyond the realm of chronic wounds and finds relevance in other medical contexts—such as magnetic tumor targeting, medical equipment sterilization, lesion localization, and more—positioning IONPs as novel auxiliary materials primed to meet the demands of clinical practice.

## Figures and Tables

**Figure 4 pharmaceutics-15-02327-f004:**
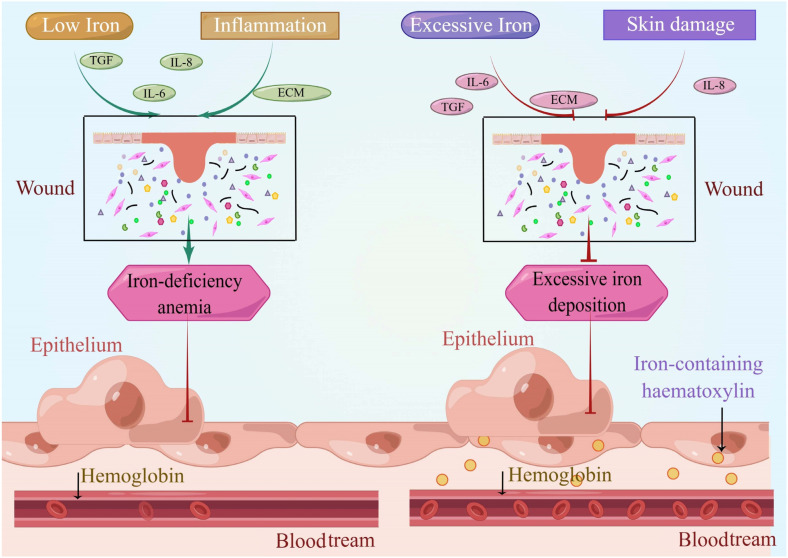
When there is a lack of iron in the tissues, it leads to iron deficiency anemia, resulting in a shortage of oxygen and nutrients in the surrounding tissues. Conversely, an iron overload in the skin and tissues leads to excessive iron deposition, characterized by the presence of the hallmark substance, ferric hemosiderin. Whether the iron levels are deficient or excessive, abnormal iron levels can lead to compromised wound healing. By conducting research within the clinical setting, we have the opportunity to shift our focus from the hospital ward to the controlled environment of the research laboratory bench. This transition allows for a comprehensive investigation into the intricate correlation between iron levels and the process of wound healing.

**Figure 7 pharmaceutics-15-02327-f007:**
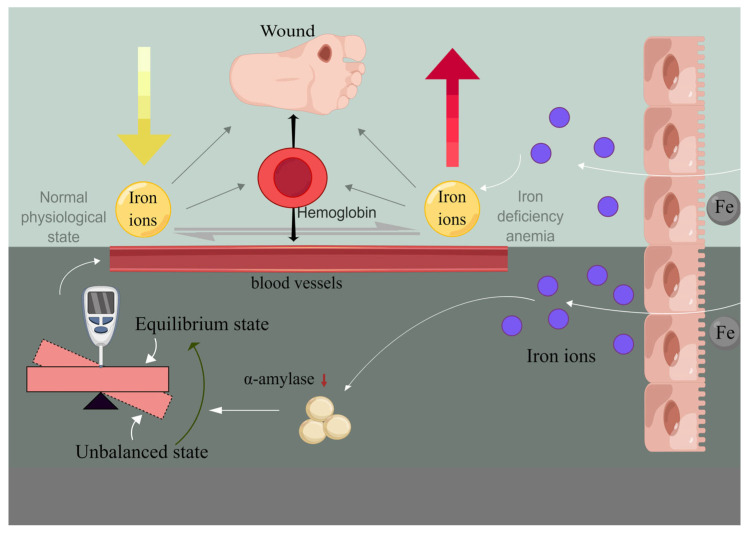
On the one hand, IONPs can be administered to the wound through dressings, facilitating the gradual release of iron ions to rectify irregular local iron concentrations within the wound. Conversely, it modulates the blood glucose levels in diabetic patients by attenuating the activity of alpha-amylase.

**Figure 9 pharmaceutics-15-02327-f009:**
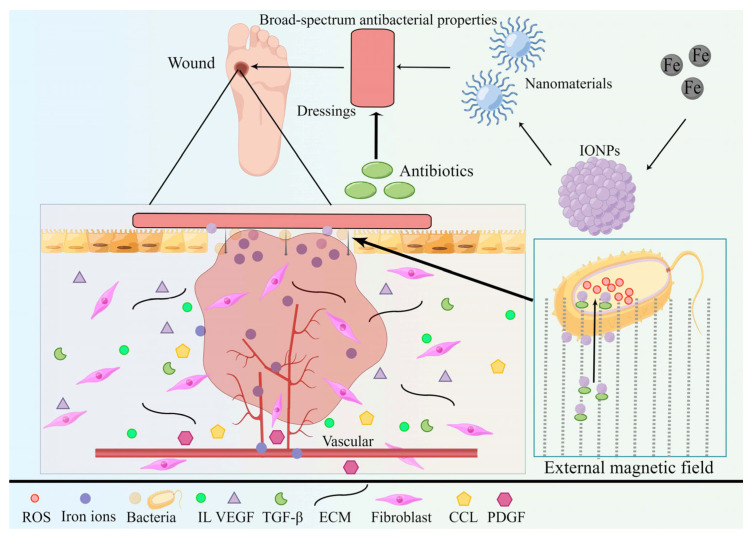
Iron NPs, when integrated with antibiotics within wound dressings, hinder bacterial infection occurrence while curtailing antibiotic volume. Notably, SPIONs can more effectively and precisely infiltrate cell membranes and walls under the influence of external magnetic fields (The arrow indicates the direction of SPION movement), effectively transporting antibiotics for enhanced sterilization.

**Table 1 pharmaceutics-15-02327-t001:** Structure and classification of common iron oxides.

Main Components	Color	Shape	Crystal Structure
α-FeOOH	Yellow	Needle-shaped	Oblique Square
β-FeOOH	Gold color	Needle-shaped	Quadrangular
γ-FeOOH		Long flakes or needle-shaped	Oblique Square
δ-FeOOH	Brown	Hexagonal flake	Hexagonal
Fe_3_O_4_	Black	Spherical or spindle-shaped	cubic shape
α-Fe_2_O_3_	Red	Spherical or spindle-shaped	Triangular
γ-Fe_2_O_3_		Needle-shaped or spindle-shaped	cubic shape

**Table 2 pharmaceutics-15-02327-t002:** Common applications of IONPs and SPIONs.

Diameter	Iron Nanoparticles	Main Roles
1–100 nm	IONPs	Catalysts [40], Pigments [41], Design of various sensors [42], removal of heavy metals from water pollution [43], antibacterial [44], treating cancer [45], UV filter, electrostatic shielding materials, etc.
<30 nm	SPIONs	imaging [49], tissue repair and cell differentiation [50], immunoassay [51], thermal therapy and drug carriers [52,53], magnetic materials, sensitive materials, etc.

**Table 3 pharmaceutics-15-02327-t003:** Partial characterization of iron oxide nanoparticles and related performance.

Characterization	Avenues	Effect	Note
Small in diameter, volume, and large in surface area-to-mass ratio	Pass through the cell wall and cell membrane	Affecting the activity of enzymes, interfering with normal cell structure	IONPs are adsorbed on bacteria through electrostatic interactions with cell membranes
oxidative stress	Release a large number of free radicals	Destroys individual components of the bacterial cell	Increase the permeability of the mitochondrial outer membrane
Carriers and mediators of antibiotics	Help antibiotics penetrate the cell membrane and cell wall	A direct bactericidal effect	Curb the misuse of antibiotics
The interaction of NPs with the cell barrier	Contact with the cell surface	Have antibacterial effects on both Gram-positive and Gram-negative bacteria	Differential antibacterial activity against Gram-negative and Gram-positive bacteria

## Data Availability

Not applicable.

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
