# Peer review of "Iron Nanoparticles Open Up New Directions for Promoting Healing in Chronic Wounds in the Context of Bacterial Infection"

_pharmaceutics, 2023, doi:10.3390/pharmaceutics15092327_

Round 1

Reviewer 1 Report

The report demonstrated interesting new approach reviewed in context with the iron NPs wound healing effect, however there are some potential concern given below in my scientific comments to improve the paper titled "Iron nanoparticles open up new directions for promoting healing in chronic wounds in the context of bacterial infection ". By addressing these comments, the paper will provide a more comprehensive and detailed account of the research, its methods, findings, and potential significance, making it more valuable to the scientific community.

Clarify Specific Iron Nanoparticle Types: In the abstract, provide a clearer differentiation between various types of iron nanoparticles (e.g., iron oxide nanoparticles, superparamagnetic iron oxide nanoparticles) to avoid confusion among readers who might not be familiar with the nuances of nanoparticle categorization.

Expand on Antibacterial Mechanisms: Elaborate on the antibacterial mechanisms of iron oxide nanoparticles. Provide more specific details on how these nanoparticles interact with bacterial cells, disrupt their membranes, and prevent bacterial growth. Additionally, discuss any potential concerns regarding the development of bacterial resistance to these nanoparticles over prolonged usage.

Include Comparative Data: Include comparative data or studies that showcase the superiority of iron nanoparticles over other wound-healing materials citing a recent study https://doi.org/10.1016/j.colsurfa.2023.131575 to make references up to date. This will strengthen the argument for the unique benefits of iron nanoparticles in promoting wound healing and combating bacterial infection.

Discuss In Vivo Studies: Incorporate findings from in vivo studies that have investigated the effectiveness of iron nanoparticles in wound healing. Mention any relevant animal model studies and their outcomes to provide more substantial evidence of the potential clinical benefits.

Address Potential Toxicity: Discuss the potential cytotoxicity and biocompatibility concerns associated with iron nanoparticles in addition to anti-anemic effect and glucose metabolic effects. Address how these concerns have been mitigated or considered in current research, and discuss strategies to ensure the safety of patients when using these nanoparticles for wound healing.

Detail External Magnetic Field Applications: When discussing superparamagnetic iron oxide nanoparticles, provide specific examples of how these nanoparticles can be manipulated using an external magnetic field to enhance wound healing. Explain the mechanism by which this manipulation occurs and its potential implications for wound treatment.

Explore Anemia and Glucose Abnormalities: Given that iron nanoparticles have properties beyond wound healing, expand on how they can contribute to the management of anemia and glucose abnormalities. Provide insights into the underlying mechanisms and existing research in this context.

Highlight Multi-Functionality: Emphasize the multi-functional nature of iron nanoparticles in wound healing. Discuss their ability to address multiple challenges simultaneously, such as antibacterial activity, acceleration of wound closure, and potential systemic health benefits. In addition, along line 277-279 on page 9, cite a recent report https://doi.org/10.3390/coatings13061095 to support the statement therein.

Address Limitations and Future Directions: Acknowledge any limitations in the current state of research on iron nanoparticles for wound healing. Discuss areas where further investigation is needed and suggest potential future research directions to encourage continued exploration in this field.

Expand Beyond Chronic Wounds: While the authors suggests the potential broader applications of iron nanoparticles beyond chronic wounds, it does not elaborate on these possibilities. Include specific examples of other medical contexts where iron nanoparticles could play a role in promoting healing or combating bacterial infection.

Moderate editing of English language required, aprticualrly abstarct needs to be rewritten with focus and clarity

Author Response

The report demonstrated interesting new approach reviewed in context with the iron NPs wound healing effect, however there are some potential concern given below in my scientific comments to improve the paper titled "Iron nanoparticles open up new directions for promoting healing in chronic wounds in the context of bacterial infection ". By addressing these comments, the paper will provide a more comprehensive and detailed account of the research, its methods, findings, and potential significance, making it more valuable to the scientific community.

(1) Clarify Specific Iron Nanoparticle Types: In the abstract, provide a clearer differentiation between various types of iron nanoparticles (e.g., iron oxide nanoparticles, superparamagnetic iron oxide nanoparticles) to avoid confusion among readers who might not be familiar with the nuances of nanoparticle categorization.

Answer: As suggested by you, we used the diameter of iron oxide nanoparticles and superparamagnetic iron oxide nanoparticles as the separating line, with 30nm as the dividing line. This allows anyone to distinguish between the two sorts of categories visually.

(2) Expand on Antibacterial Mechanisms: Elaborate on the antibacterial mechanisms of iron oxide nanoparticles. Provide more specific details on how these nanoparticles interact with bacterial cells, disrupt their membranes, and prevent bacterial growth. Additionally, discuss any potential concerns regarding the development of bacterial resistance to these nanoparticles over prolonged usage.

Answer: First, we suggested that IONPs can better adsorb to bacterial surfaces by their physical properties (e.g., size) in 3.1.3 and cited [45]: doi:10.3390/ijms140918110.,[46]: doi:10.1016/j.jconrel.2011.07.002. as an argument. By changing the membrane structure and signal transduction, IONPs can also stop the growth and proliferation of bacteria. On the basis that the integrity of the shell is further damaged, they utilize their physical properties and electrostatic attraction to enhance barrier permeability ([88] doi:10.1021/es060999b,[89] doi:10.1021/nn203785a.). Mechanisms of drug resistance include the inactivation of antimicrobial drugs by bacterial production of enzymes and the alteration of bacterial outer membrane permeability. IONPs can counteract the formation of drug resistance by increasing permeability, transporting antibiotics, and penetrating biofilms, which can delay and reduce the process of acquired resistance formation as much as possible ([98] doi:10.3389/fbioe.2019.00141.).

(3) Include Comparative Data: Include comparative data or studies that showcase the superiority of iron nanoparticles over other wound-healing materials citing a recent study https://doi.org/10.1016/j.colsurfa.2023.131575 to make references up to date. This will strengthen the argument for the unique benefits of iron nanoparticles in promoting wound healing and combating bacterial infection.

Answer: In conjunction with your comments, we have cited a recent study https://doi.org/10.1016/j.colsurfa.2023.131575 to make references up to date. Compared to other wound healing materials, IONPs take a holistic approach to treating chronic wounds.

(4) Discuss In Vivo Studies: Incorporate findings from in vivo studies that have investigated the effectiveness of iron nanoparticles in wound healing. Mention any relevant animal model studies and their outcomes to provide more substantial evidence of the potential clinical benefits.

Answer: Mention any relevant animal model studies and their outcome, for example, (SPIONs) significantly reduced blood glucose levels in diabetic rats (situated at. [81] doi:10.1016/j.lfs.2020.117361.), intravenous injection experiments in mice of Mario Ledda, et al verified the biocompatibility of IONPs.(situated at 3. [82] doi:10.1039/c9nr09683c.), Guo J. et al established an animal model of Staphylococcus aureus infection in wound mice which affirmed the in vivo antibacterial effect of IONPs(situated at 3.1.1 [85] doi:10.1093/rb/rbac041.), etc.

(5) Address Potential Toxicity: Discuss the potential cytotoxicity and biocompatibility concerns associated with iron nanoparticles in addition to anti-anemic effect and glucose metabolic effects. Address how these concerns have been mitigated or considered in current research, and discuss strategies to ensure the safety of patients when using these nanoparticles for wound healing.

Answer: In 5.2, we focused on the potential cytotoxicity and biocompatibility issues associated with iron nanoparticles. SPIONs with a particle size of less than 2 nm exhibit potential toxic effects([120] doi:10.2147/ijn.S30320.), Iron overload causes structural damage and dysfunction. ([122] doi:10.3238/arztebl.m2021.0290.). We propose a three-pronged approach to control biotoxicity and ensure patient safety: material preparation, patient screening, and prognostic monitoring.

(6) Detail External Magnetic Field Applications: When discussing superparamagnetic iron oxide nanoparticles, provide specific examples of how these nanoparticles can be manipulated using an external magnetic field to enhance wound healing. Explain the mechanism by which this manipulation occurs and its potential implications for wound treatment.

Answer: We explain in detail in 3.2.1 how SIONPs can be modulated by external magnetic fields to promote wound healing. The position and trajectory of the IONPs are controlled by magnetic field gradients, and different combinations of magnetic field gradients can be used to more precisely adjust the targeting of the SIONPs([106] doi:10.1098/rsif.2022.0576.).In addition, external coatings can be added to improve further the stability and functionality of SIONPs ([107] doi:10.3390/antibiotics10091138. ,[108] doi: 10.1016/j.saa.2014.07.059.).

(7) Explore Anemia and Glucose Abnormalities: Given that iron nanoparticles have properties beyond wound healing, expand on how they can contribute to the management of anemia and glucose abnormalities. Provide insights into the underlying mechanisms and existing research in this context.

Answer: In 3.1.2, we linked anemia and blood glucose abnormalities to the chronic wounds mentioned earlier. The regulations of anemia and glycemic abnormalities by IONPs are primarily mediated through iron release and α-amylase inhibitory activity([94] doi:10.2147/IJN.S121956.).We also argued this point by citing animal modeling experiments ([81] doi: 10.1016/j.lfs.2020.117361.). In addition, the slow and sustained release of iron effectively avoids iron overload and potential biotoxicity ([2] doi:10.3390/antibiotics10050524.).

(8) Highlight Multi-Functionality: Emphasize the multi-functional nature of iron nanoparticles in wound healing. Discuss their ability to address multiple challenges simultaneously, such as antibacterial activity, acceleration of wound closure, and potential systemic health benefits. In addition, along line 277-279 on page 9, cite a recent report https://doi.org/10.3390/coatings13061095 to support the statement therein.

Answer: In 3.2.2, we elaborated on the multi-functional nature of iron nanoparticles in wound healing. Not only can it improve the magnetic properties, but it can also influence the behavior of nanoparticles in vivo. The main aspects include acting as a drug carrier to destroy cells and fully utilizing the effective concentration of the drug([109] doi:10.1016/j.addr.2008.03.018.).Based on your comments, We cite a recent report https://doi.org/10.3390/coatings13061095 to support the statement therein ([78] doi:10.3390/coatings13061095.).

(9) Address Limitations and Future Directions: Acknowledge any limitations in the current state of research on iron nanoparticles for wound healing. Discuss areas where further investigation is needed and suggest potential future research directions to encourage continued exploration in this field.

Answer: We have added a new 5.3 to 5. to discuss the limitations you raised. The two main directions include standardized preparation and external magnetic fields. The characterization and standardization of SPIONs need to be further explored and practiced ([109] [109] doi:10.1016/j.addr.2008.03.018.).If SPIONs are to achieve more precise and diversified localization and trajectories, complex magnetic field gradients are required, which is a relatively high demand for hardware facilities at this stage of the research ([120] doi:10.2147/ijn.S30320.).

(10) Expand Beyond Chronic Wounds: While the authors suggests the potential broader applications of iron nanoparticles beyond chronic wounds, it does not elaborate on these possibilities. Include specific examples of other medical contexts where iron nanoparticles could play a role in promoting healing or combating bacterial infection.

Answer: At the end of the manuscript, we suggested that the evolution of IONPs and external magnetic fields may bring new adjustments in clinical dosing regimens. We can propose visions in more ways, such as magnetic targeting of tumors, sterilization of medical supplies and machinery, localization of lesions, etc.

These are the improvements we've made to the manuscript based on your comments. In addition to this, we have made English revisions to the articles. With your comments and suggestions, we are confident that our manuscript will become more scientifically valuable. Thank you again for your patient review and attentive guidance.

Reviewer 2 Report

The authors give a fairly extensive review of the use of iron particles in wound healing. Overall, the paper is well written. My main concern is that while the authors briefly mention potential concerns of using iron to control infection in their "Limitations", iron is also utilized by bacteria for growth and potentially could be a problem. I feel that there should be more discussed about the potential problems of using iron for infected wounds. The body has developed a very extensive iron scavenging system (transferrin, etc.) that limit iron's availability. Most of the discussion in the paper talk about hemoglobin and hypoxia, which is really not a free iron issue. 

Author Response

The authors give a fairly extensive review of the use of iron particles in wound healing. Overall, the paper is well written. My main concern is that while the authors briefly mention potential concerns of using iron to control infection in their "Limitations", iron is also utilized by bacteria for growth and potentially could be a problem. I feel that there should be more discussed about the potential problems of using iron for infected wounds. The body has developed a very extensive iron scavenging system (transferrin, etc.) that limit iron's availability. Most of the discussion in the paper talk about hemoglobin and hypoxia, which is really not a free iron issue. 

Answer: First of all, thank you very much for recognizing our manuscript. It is our honor to have your guidance and review. We believe that our manuscript will be better and more scientifically valuable after your review and suggestions.

Regarding your concern that iron favors bacterial growth, we may not have described it enough in the original manuscript. With modifications, we have discussed the slow release of iron through IONPs. The slow release of iron can inhibit bacterial growth, and we believe that the unique physical properties of IONPs themselves can sufficiently reduce this risk. We suggested that IONPs can better adsorb to bacterial surfaces by their physical properties (e.g., size) in 3.1.3 and cited [45]: doi:10.3390/ijms140918110.,[46]: doi:10.1016/j.jconrel.2011.07.002. as an argument. By changing the membrane structure and signal transduction, IONPs can also stop the growth and proliferation of bacteria. On the basis that the integrity of the shell is further damaged, they utilize their physical properties and electrostatic attraction to enhance barrier permeability([88] doi:10.1021/es060999b,[89] doi:10.1021/nn203785a.). Mechanisms of drug resistance include the inactivation of antimicrobial drugs by bacterial production of enzymes and the alteration of bacterial outer membrane permeability. IONPs can counteract the formation of drug resistance by increasing permeability, transporting antibiotics, and penetrating biofilms, which can delay and reduce the process of acquired resistance formation as much as possible ([98] doi:10.3389/fbioe.2019.00141.).

Based on your comments, we are discussing the potential problems of iron treatment of wounds more in our manuscript revision. In 5.2, we focused on the potential cytotoxicity and biocompatibility issues associated with iron nanoparticles. SPIONs with a particle size of less than 2 nm exhibit potential toxic effects([120] doi:10.2147/ijn.S30320.), Iron overload causes structural damage and dysfunction. ([122] doi:10.3238/arztebl.m2021.0290.). We propose a three-pronged approach to control biotoxicity and ensure patient safety: material preparation, patient screening, and prognostic monitoring.

In addition, the principal purpose of our discussion of hemoglobin and hypoxia in this article is to introduce the relationship between iron and the organism. Iron is not only contained in wounds but also closely related to the organism as a whole. The release of iron ions is an advantage of IONPs, based on your proposal of an iron scavenging system (transferrin, etc.), which we will discuss in a follow-up study.

These are some of the changes we have made, you can see the full document for details. Thanks again for reviewing.

Round 2

Reviewer 1 Report

accept

Reviewer 2 Report

Accept